# Active particles bound by information flows

Utsab Khadka[1], Viktor Holubec [2,3], Haw Yang [1] & Frank Cichos [4]

Self-organization is the generation of order out of local interactions. It is deeply connected to many fields of science from physics, chemistry to biology, all based on physical interactions. The emergence of collective animal behavior is the result of self-organization processes as well, though they involve abstract interactions arising from sensory inputs, information processing, storage, and feedback. Resulting collective behaviors are found, for example, in crowds of people, flocks of birds, and swarms of bacteria. Here we introduce interactions between active microparticles which are based on the information about other particle positions. A real-time feedback of multiple active particle positions is the information source for the propulsion direction of these particles. The emerging structures require continuous information flows. They reveal frustrated geometries due to confinement to two dimensions and internal dynamical degrees of freedom that are reminiscent of physically bound systems, though they exist only as nonequilibrium structures.

[1] Department of Chemistry, Princeton University, Princeton, NJ 08544, USA. [2] Institute for Theoretical Physics, Universität Leipzig, 04103 Leipzig, Germany. [3] Department of Macromolecular Physics, Faculty of Mathematics and Physics, Charles University, V Holešovičkách 2, CZ-180 00 Praha, Czech Republic. [4] Peter Debye Institute for Soft Matter Physics, Universität Leipzig, 04103 Leipzig, Germany. Correspondence and requests for materials should be addressed to F.C. (email: cichos@physik.uni-leipzig.de)

Active particles serve as simple microscopic model systems for living objects such as birds, fish, or people and mimic in particular the propulsion of bacteria or cells without the complexity of physical properties and chemical networks in living objects[1]. They consume energy to propel persistently and as such they have given considerable insight into collective behaviors of active materials already[2–6]. With their bare function of self-propulsion they are, however, missing the important ingredients of sensing and feedback, which most living objects from cells up to whole organisms have in common. All of their living relatives have signaling inputs which they use to gain information about the environment. Employing external information, organisms interact such that birds and fish are able to self-organize into flocks or schools[7–9] and, on a microscopic level, cells may regulate gene expression[10,11]. Using sensing, information processing and feedback, living systems may go beyond what is prescribed by physical interactions such as the Coulomb, van der Waals or hydrophobic interactions. For cells/bacteria the information about cellular density (quorum sensing) is, for example, inferred from the concentration of signaling molecules released by the cells leading to a regulation of various physiological activities such as biofilm formation[10]. Concentration gradients of nutrients may serve as sensory input leading to a directed motion termed chemotaxis[12]. For birds, the information used for the formation of flocks is suggested to be the visual perception of the number of neighboring birds[8]. In this respect all information that is processed by the organism is linked to a physical representation[13] (e.g. concentration, number of objects, orientation,…), but the resulting structure and dynamics is disconnected from a direct physical interaction. In the particular examples mentioned above, though, the structure formation depends on the active response of the organism and its ability to steer based on its recognition of the environment[7,14–16]. While active particles do not have sensory inputs, information processing units and feedback mechanisms built in yet, suitable control mechanisms may introduce this complexity fostering the exploration of new emergent phenomena. An information exchange between active particles has not been tackled so far, but seems to be a natural step towards extending their functionality.

Like bacteria, active particles have to break the time symmetry of low Reynolds number hydrodynamics in order to propel. They have to provide asymmetries to generate directed motion. In many cases this asymmetry is built into the structure in form of two hemispheres with different chemical or physical properties, so called Janus particles[1,17]. As the propulsion direction is bound to the symmetry axis of the particle, rotational diffusion becomes a relevant process that limits the control of Janus particles[18,19]. The self-propulsion mechanism presented below relies on a scheme for generating self-thermophoresis[20]; unlike past approaches our new scheme utilizes the spatially controlled asymmetric input and release of energy around a symmetric particle. This scheme delivers a precise control of each individual particle in a larger ensemble, and allows us to demonstrate how active particles may form structures just by the exchange of information on other particle positions using a feedback control mechanism for steering the particles.

## Results

**Particle control**. Our active particle is constructed of a melamine resin sphere with 30% of the surface uniformly decorated by gold nanoparticles of about 10 nm diameter (Supplementary Fig. 1). Illuminating the particle asymmetrically by a deflectable focused laser beam generates an inhomogeneous surface temperature and results in the desired self-thermophoretic motion based on thermo-osmotic surface flows[21,22] (Fig. 1a). This design allows for

a new control scheme for active particle steering. To control the particle propulsion direction, we place the laser beam's focal spot near the circumference of the particle. The propulsion direction is then the vector from that heated circumferential spot to the particle center. Different from standard active particles, the timescale of the rotational diffusion of the particle is irrelevant due to the missing particle asymmetry (Supplementary Fig. 5).

We first evaluate properties of a single active particle and the control accuracy in an experiment driving the particle between two target positions and confining it for a certain time at the targets (Fig. 1b). We extract the dependence of the active particle propulsion velocity on the heating power from this experiment finding a non-linear scaling (Fig. 1c) as the particle slips away from the focus during the camera exposure time. Incorporating this slipping process in a model (Supplementary Note 1) reproduces this nonlinear increase qualitatively (Supplementary Fig. 4). The confinement at the target position can be characterized by a positioning error $\sigma = \sqrt{\langle (\mathbf{r} - \mathbf{r}_t)^2 \rangle}$ – the standard deviation of the probability density of the particle–target distance in the steady state. Here, $\mathbf{r}$ and $\mathbf{r}_t$ are the coordinates of the swimmer and target, respectively. The positioning error obeys two regimes[18,19,23,24]. When the displacement of the particle due to the propulsion is smaller than the diffusive displacement during the exposure time $\Delta t_{exp}$, the positioning error is reflected by a simple sedimentation model. A constant particle speed $v_{th}$ drives the particle radially towards the target position against the Brownian motion with a diffusion coefficient $D_0$. Hence, the density distribution in the steady state is exponential with a characteristic length scale (the sedimentation length in two dimensions) $\rho = \sqrt{6 D_0 / v_{th}}$ as indicated by the red dashed curve in Fig. 1d[19]. When the active particle speed increases, an overshooting of the particle over the target position due to the finite sampling of the position of the particle with the camera exposure time $\Delta t_{exp}$ defines the positioning error[25]. The overshooting distance equals the traveled distance within the time between two frames $\Delta t_{exp}$ and increases linearly with the velocity of the active particle as shown by the black dashed line. The sum of both contributions determines the positioning error $\sigma$ which is depicted for different particle speeds from the experiments (triangle markers, forward and backward motion in Fig. 1b) together with the predicted curve $\sigma = \sqrt{6 D_0^2 / c^2 v_{th}^2 + c^2 v_{th}^2 \Delta t_{exp}^2}$ with no free parameters (solid black curve). A minimum positioning error of $\sigma = 370$ nm is found for the exposure time of $\Delta t_{exp} = 80$ ms, a diffusion coefficient of $D_0 = 0.23$ μm² s⁻¹ (experimentally found as compared to theory $D_0 = 0.20$ μm² s⁻¹) and a velocity of $v_{th} = 1.3$ μm s⁻¹. A factor $c = 2$ accounts for the fact that the particle travels twice the distance due to the feedback delay of one exposure time $\Delta t_{exp}$.

**Multiple active particles, swarms and structures**. Multiple particle control is introduced by illuminating multiple particles at suitable positions at their respective circumferences. In the current setup, an acousto-optic deflector multiplexes the focused heating laser spot between different particle positions within one exposure of duration $\Delta t_{exp}$ (Methods). The incident heating power is therefore available for a time $\Delta t_{exp}/N$ to each of the $N$ particles and the average heating power per particle decreases when keeping the overall incident laser power constant. Figure 2a depicts the control of six individual active particles in a spatially fixed pattern of six target positions, arranged as the nodes of a symmetric hexagon (Supplementary Movie 2). The particles were initially distributed randomly in the field of view. Once the control was initiated, each of the particles was first driven towards its nearest target, after which it was confined there. The resulting

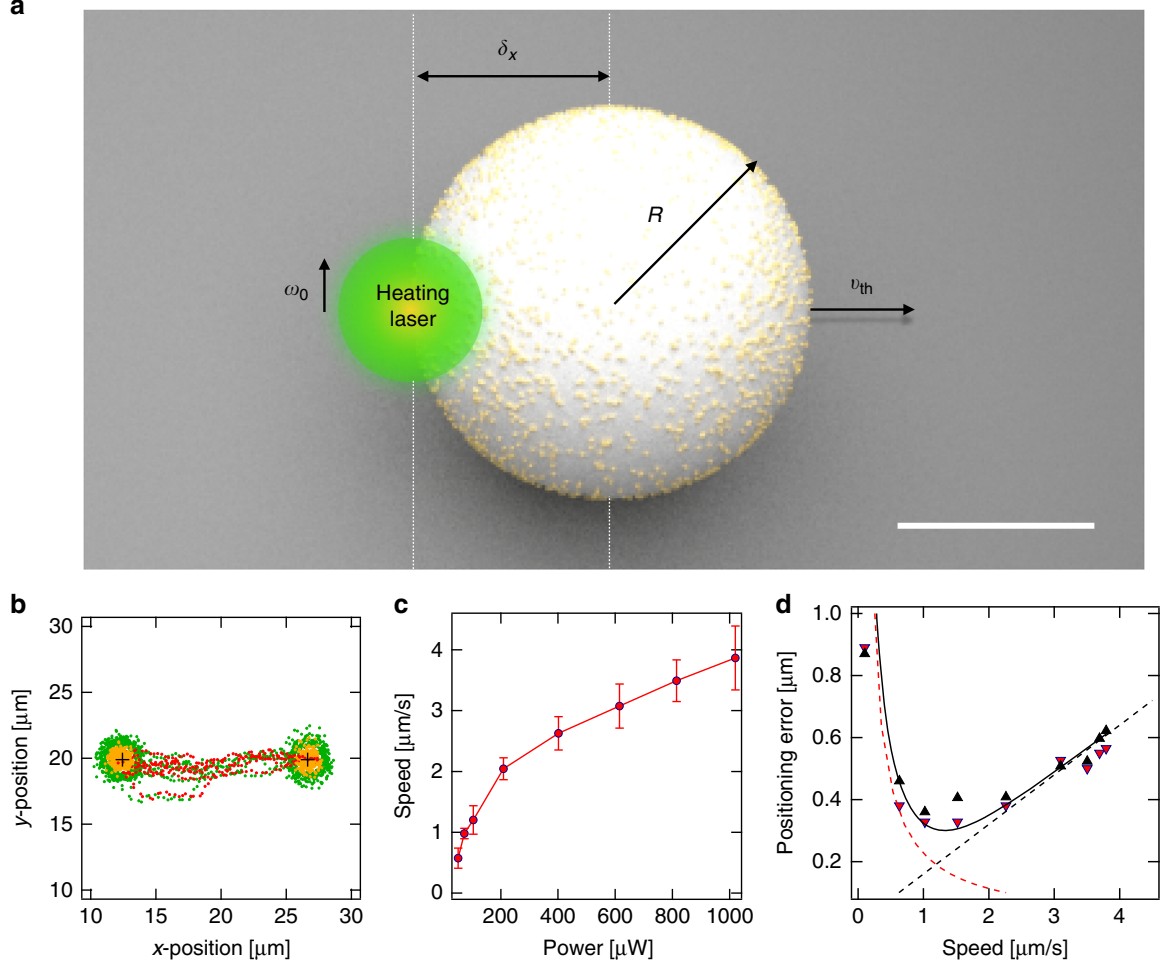

**Fig. 1** Symmetric self-thermophoretic active particle and its properties. **a** Scheme of the self-thermophoretic active particle composed of a melamine resin particle ($R = 1.09 \, \mu m$) covered at 30% surface area by 10 nm gold nanoparticles. The nanoparticles can be heated by an incident laser beam (beam waist $\omega_0$) to generate an inhomogeneous temperature profile along the particle surface. This profile causes a thermo-osmotic slip flow propelling the particle at $v_{th}$ away from the laser. The velocity of the particle is determined by the displacement $\delta x$ of the laser focus from the particle center. The scale bar has a length of $1 \, \mu m$. **b** The particle velocity and control accuracy is derived from an experiment driving the particle between two target positions and confining it for 100 frames at each position. Example trajectory points (laser positions (green), particle position during trapping (orange), particle positions during driving (red)) (see Supplementary Movie 1). **c** Extracted particle speed as a function of the heating power. The error bars reflect the standard deviations from at least driving periods as shown in Supplementary Fig. 5. The nonlinear dependence is analyzed in the Supplementary Information. **d** Control accuracy as a function of the particle speed determined from the experiment in Fig. 1b. The upwards and downwards triangles correspond to the data from the left and right target, respectively. The dashed lines are the contributions from the 2-dimensional sedimentation (red) and the particle overshooting (black). The solid black curve represents the sum of both contributions

steady state distribution of the particles overlaps with the distribution of the assigned target coordinates, as shown in Fig. 2b. The accuracy of the particle control follows the dependencies revealed earlier in Fig. 1. In contrast to optical tweezers, the applied scheme does not involve external forces, but just dissipative fluxes and is thus physically different from common trapping[24]. An active particle always dissipates energy even if it appears stationary, i.e., in a collision with a wall, while this is not the case for a particle driven by external forces through a liquid. Yet, the stationary position distribution of each particle around its target may be realized by external forces[25,26] and appears in the example of Fig. 2 to be similar to what has been achieved with the help of holographic optical tweezers[27,28].

A collection of particles may be driven to a target either by first arranging them into a structure and subsequently translocating the structure, or by driving each particle directly to the target without prior structural arrangement, as demonstrated in the Supplementary Movies. While the former approach results in a

collective motion resembling the transport of an organized fleet of vehicles (see Supplementary Movie 2), in the latter case the unstructured collective driving results in a swarm-like motion determined by the active particles steric repulsion, Brownian motion and the propulsion towards the target (see Fig. 2c). In addition to the switching of propulsion directions, the local modulation of the propulsion speed may also lead to a well-controlled effective potential sculpting the particle probability density distributions according to $p(\mathbf{r}) \propto 1/v_{th}(\mathbf{r})$ much like in the motility-induced phase transitions observed in active particle ensembles[29,30].

**Self-organized active particle molecules**. The described multiple particle control scheme opens the realm of feedback-induced virtual interactions for active particles. Here, particles are allowed to exchange information via the real-time tracking feedback loop of the microscopy system. It is similar to a situation where birds

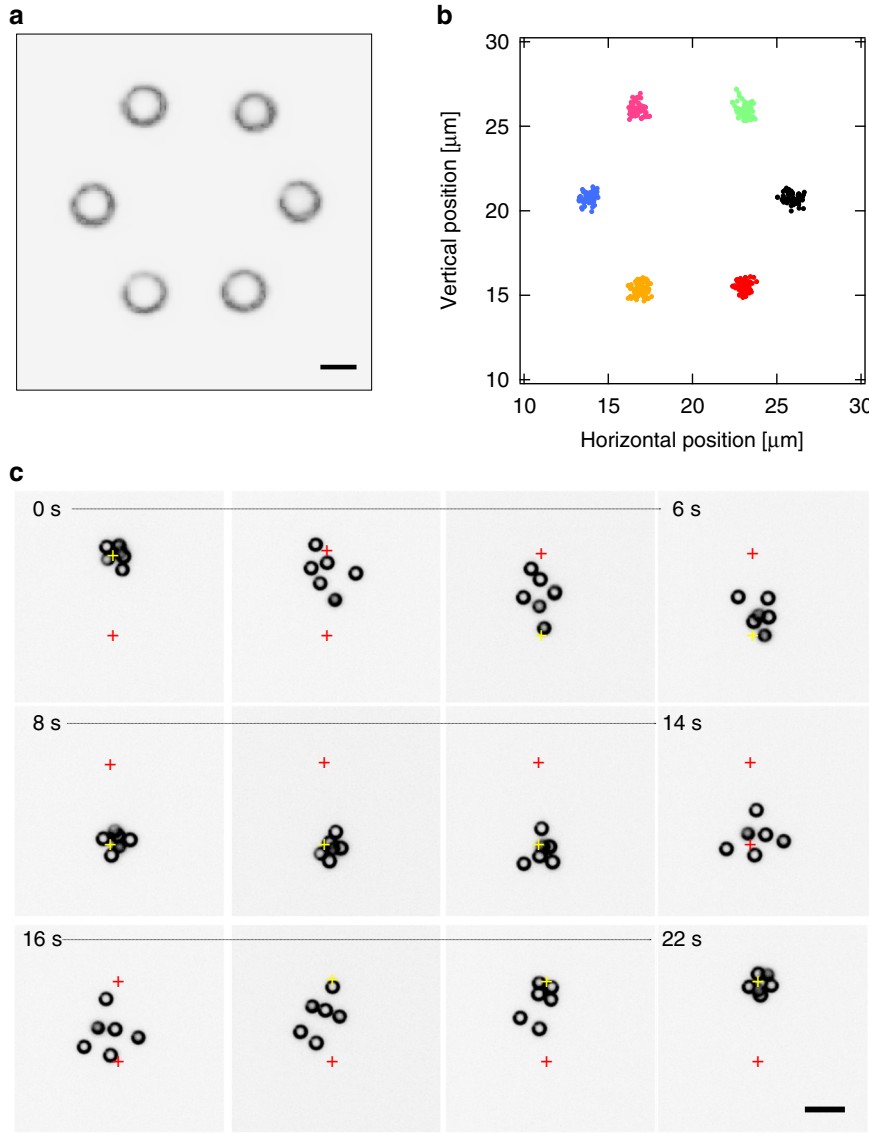

**Fig. 2** Controlling multiple self-thermophoretic active particles. **a** Darkfield microscopy snapshot (inverted grayscale) of six active particles ($R = 1.09\,\mu m$) arranged at the nodes of a symmetric hexagon ($7.1\,\mu m$ edge length) with the help of the particle control procedure described in the text. The incident laser power per particle is $P = 0.2\,mW$. The scale bar has a length of $2\,\mu m$. **b** Corresponding trajectory points of the particles over a time period of 11 s with $\Delta t_{exp} = 110\,ms$ exposure time/inverse frame rate. **c** Dark-field microscopy image series (inverted grayscale) of six active particles driven between two target positions separated by $14.2\,\mu m$. Targets are colored for clarity. The incident heating power per particle is $P = 0.2\,mW$ and the time resolution of the experiment is $\Delta t_{exp} = 80\,ms$. The scale bar has a length of $7\,\mu m$

or fish react to the action of their neighbors to form flocks and schools or to escape predators. It introduces a signaling channel between the particles, which can be tweaked almost arbitrarily to design virtual interactions and paves the way for a vast amount of studies from the self-organization of new structures to the information flow in flocks[9], or the application of machine learning to study adaption and the emergence of collective patterns[31]. Moreover, it provides a minimal scalable robotic system with a simple propulsion and intrinsic noise due to Brownian motion. Here, we demonstrate the structure formation by defining a pairwise control, which intends to keep the active particles at a prescribed separation distance $r_{eq}$ by just changing their propulsion direction, but not the speed. If the in-plane distance $r_{ij}$ between two particles ($i$ and $j$) is below the separation distance $r_{eq}$, the particles are pushed away from each other, each with a speed $v_{th}$. In the case $r_{ij} > r_{eq}$, the particles are pushed towards each other with the same speed, which results in an

effective V-shaped interaction potential for a pair of active particles. For a number of $N$ interacting particles, this feedback rule is represented for particle $i$ by the velocity

$$\mathbf{v}_i(t) = -v_{th}\,\mathbf{e}_i(t), \tag{1}$$

where the propulsion direction is determined by

$$\mathbf{e}_i(t) = \frac{\sum_{j \neq i}^{N} \text{sign}\left(r_{ij}(t - \delta t) - r_{eq}\right)\mathbf{e}_{ij}}{\left|\sum_{j \neq i}^{N} \text{sign}\left(r_{ij}(t - \delta t) - r_{eq}\right)\mathbf{e}_{ij}\right|} \tag{2}$$

with $r_{ij} = |\mathbf{r}_j(t - \delta t) - \mathbf{r}_i(t - \delta t)|$ and $\mathbf{e}_{ij} = (\mathbf{r}_j(t - \delta t) - \mathbf{r}_i(t - \delta t))/r_{ij}$. As stated by Eq. (1), the speed of motion is always $v_{th}$, while its direction $\mathbf{e}_i(t)$ is defined by the positions of the other active

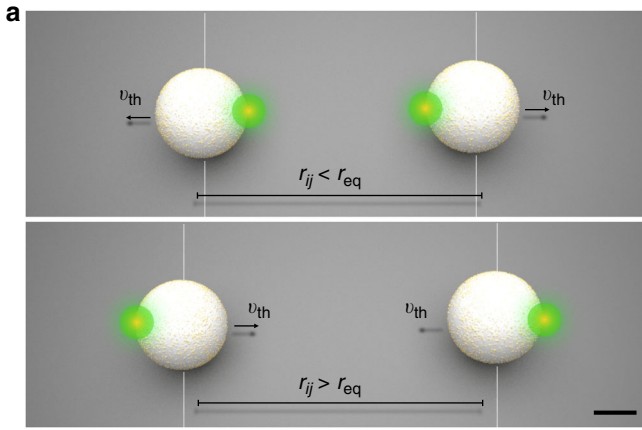

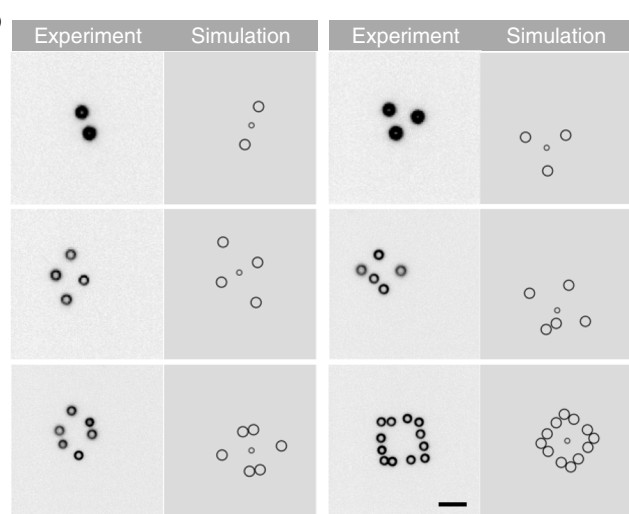

**Fig. 3** Feedback rule and self-organized active particle structures. **a** Pair interaction rule used to "bind" the active particles to each other. Particles are propelled with a speed $v_{th}$. If the distance is below $r_{eq}$, the particles are moved away from each other, if above, they are propelled towards each other. The scale bar has a length of 1 μm. **b** Example snapshots of 6 different "active particle molecules" that are bound by the interaction rule in **a**. The structures are highly dynamic (see Supplementary Movies 4-9). The scale bar has a length of 7 μm. The experimental images are compared to snapshots of corresponding numerical simulations involving Brownian motion and a simple delayed feedback as in the experiment (not drawn to scale, see Supplementary Note 2). The smaller central circle marks the center of mass. An effective potential description (not included) also reveals equivalent structures up to the pentamer

particles. The interaction rule is implemented in a real-time particle tracking loop (Methods) detecting the center positions of the active particles and steering the heating laser to the corresponding spots at their circumferences. Position measurements and action are separated by one exposure time $\Delta t_{exp}$ introducing the feedback delay $\delta t = \Delta t_{exp}$.

After a short self-organization phase, the particles arrange into freely diffusing dynamic structures that are reminiscent of simple molecules (see Supplementary Movies 4–9). They are not prescribed by a given potential energy landscape, but arrange due to the defined rules. Figure 3 depicts snapshots of 6 self-organized structures all with $r_{eq} = 7.1$ μm. Active particle molecules with $N = 2$ and $N = 3$ particles (dimer, Fig. 4 and trimer, Fig. 5a) form structures where all average interparticle distances correspond to the adjusted value of $r_{eq}$ (measured $\langle r_{ij} \rangle = 7.23$ μm). All clusters with a larger number of particles ($N \geq 4$)

are structurally frustrated due to the confinement in two dimensions. Four active particles cannot form a tetrahedral structure with equilateral triangular faces. Instead a 2-dimensional structure with two "isomers" is found (Fig. 5b, shown together with a timetrace of the isomerization process in Fig. 5c). Larger active particle molecules ($N > 4$) form frustrated structures with interparticle separations of less than the defined value $r_{eq}$. Figure 3b highlights snapshots of a pentamer (middle right), a hexamer (bottom left) and a dodecamer (bottom right). The dodecamer, for example, forms a transient square structure out of 4 right angle subunits of 3 particles.

A detailed picture on the experimentally observed dynamics of the active particle structures is obtained from a principle component analysis (PCA)[32] of the displacement vectors of the individual particles between successive frames in the center of mass frame $\Delta \vec{r}_i^{COM}$. The identified eigenvectors correspond to a set of $2N - 2$ orthogonal directions with the largest displacement variances in the COM frame. Figure 4b presents the two modes for the dimer structure (stretch and rotation, $N = 2$) and Fig. 5a the four modes of the trimer (symmetric stretch, bending mode, asymmetric stretch, and rotation, $N = 3$) where the modes agree well with the normal modes of an equilateral triangular structure in two dimensions. The appearance of a rotational mode is at first glance surprising as the feedback is designed to act along the connecting line between the bound particles. Due to the feedback delay and Brownian motion, the laser heating position along the circumference is fluctuating as well and introducing a coupling between the translational and rotational motion. The oscillatory motion as indicated for the two modes of the dimer (Fig. 4a) is a fundamental feature of a time-delayed negative feedback system and inherent to electronic oscillators, but also appearing at all levels of biological systems[33]. In our active particle assemblies the dynamics is controlled by three parameters, the feedback delay $\delta t$, the propulsion speed $v_{th}$, and the single particle diffusion coefficient $D_0$. Using these parameters and restricting the analysis to the dynamics of the dimer along the connecting line (stretch mode), we can model the stretch mode dynamics with an overdamped Langevin description,

$$\dot{r}_{12}(t) = -2v_{th} \, \text{sign}\left(r_{12}(t - \delta t) - r_{eq}\right) + \sqrt{4D_0}\eta_{12}(t). \quad (3)$$

Here $\eta_{12}(t)$ is a zero-mean, unit-variance Gaussian white noise, i.e., $\langle \eta_{12}(t) \rangle = 0$ and $\langle \eta(t)\eta(t') \rangle = \delta(t - t')$, such that the variance of the noise term in Eq. (3) corresponds to $4D_0$. As the bond length involves the relative motion of two particles, the relative velocity $2v_{th}$ and the relative diffusion coefficient $2D_0$ enter the equation. Its solution yields the observed oscillatory motion with a triangular shape and an oscillation period of $T = 4\delta t$ (Supplementary Note 3). Accordingly, the oscillatory dynamics vanishes for zero feedback delay and stiff structures appear. The amplitude of the motion linearly depends on the delay and the active particle velocity ($\Delta r_{12} = 2v_{th}\delta t$). A feedback delay of $\delta t = 0.11$ s and a propulsion velocity $v_{th} = 3.4$ μm s$^{-1}$ ($P_{heat} = 0.75$ mW per particle) delivers $T = 0.44$ s and $\Delta r_{12} = 0.75$ μm, which compares well to the experimental data for the dimer $T_{exp} = 0.44$ s and the amplitude of the stretch mode $\sqrt{\langle A_0^2 \rangle} = 0.72$ μm obtained from the first eigenvalue of the PCA, $A_0$. The effect of the last term in Eq. (3) is to introduce phase and amplitude noise to the oscillatory motion. The oscillations are therefore losing coherence and the autocorrelation $C_i(t) = \langle A_i(\tau) A_i(\tau + t) \rangle_\tau / \langle A_i(\tau)^2 \rangle_\tau$ of the oscillating modes $A_i(t)$ decays. The timescale of this damping is the dephasing time, which is termed $T_2$ in molecular spectroscopy. Using an approximate solution of Eq. (3) for the dimer (see Supplementary Note 3), we find a dephasing time $T_2 \approx 32\Delta r_{12}^2 \pi^{-4} D_0^{-1}$, which scales inversely with the strength of the noise given by the diffusion coefficient (see

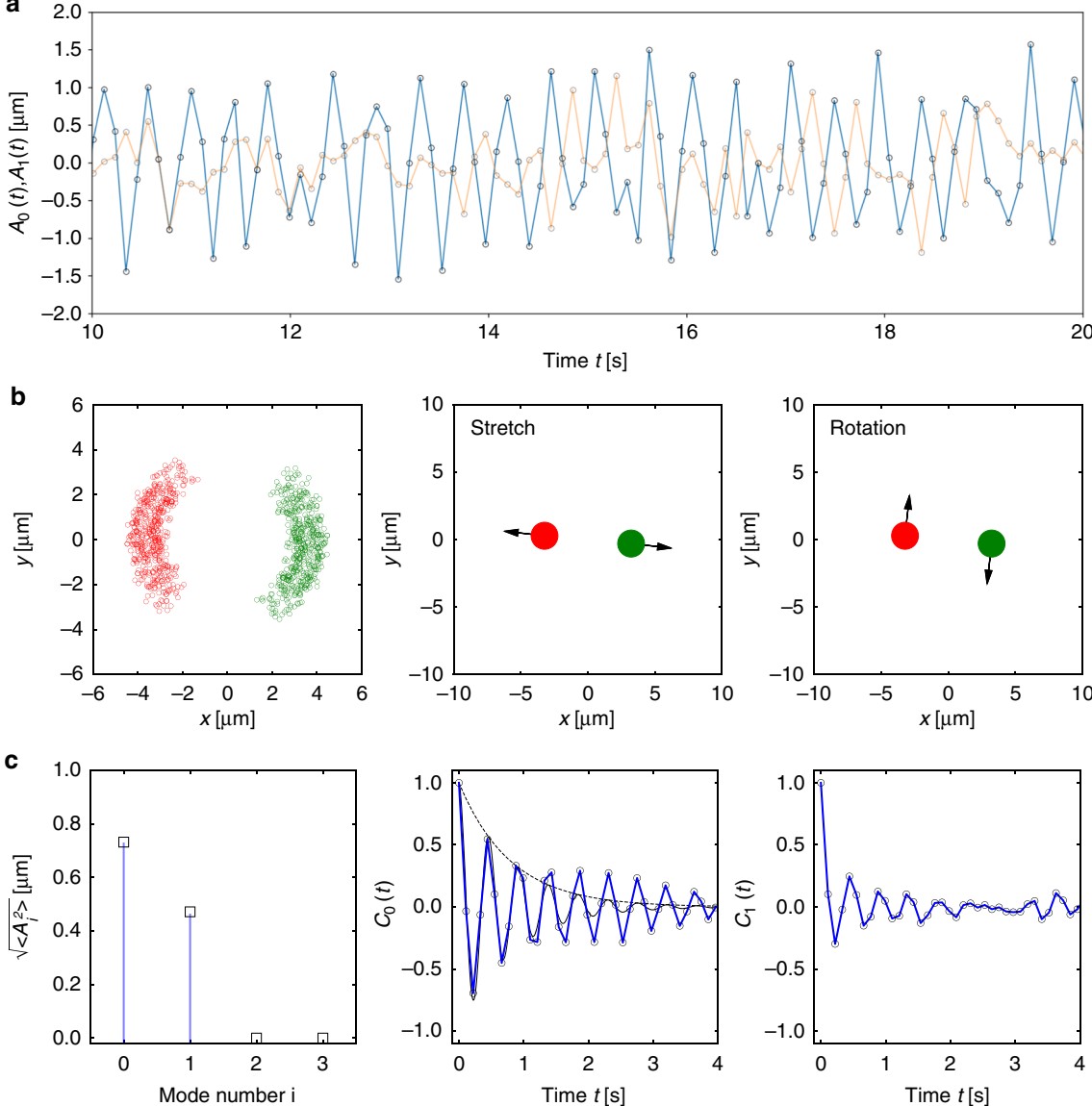

**Fig. 4** Dynamics of a self-organized active particle dimer. **a** Change of the dimer bond length parallel (blue) and perpendicular (orange) to the connecting line of the two particles. The dimer bond length is oscillating with a triangular shaped elongation with a period of $T = 0.44$ s. The period corresponds to four times the feedback delay time $\delta t = 0.11$ s. **b** Left: trajectory points of the two bound particles in the center of mass (COM) frame. Middle and right: principle components of the particle displacements in the center of mass frame as obtained from a principle component analysis (PCA). **c** Left: principle component amplitudes as calculated for the two modes in **b**. Middle and right: autocorrelation functions for the displacements of the particles in the center of mass frame of the two eigenvectors obtained from the PCA. The two modes reveal a damped oscillation due to the Brownian motion of the active particles. The middle graph shows in addition the theoretical prediction for the oscillation (black solid line) and the exponential decay due to the dephasing (black dashed line)

Supplementary Note 3). For the dimer bond length oscillation displayed in Fig. 4c we find a dephasing time of $T_2 = 0.8$ s (dashed line). In larger structures, each particle contributes to the total noise such that coherent oscillations as observed for the dimer disappear quickly with growing size of the cluster.

## Discussion

The presented symmetric active particles and the introduced manipulation technique allow us to self-assemble structures by designed feedback controlled interaction rules. Similar structures of passive colloidal particles have been assembled in previous work with the help of external forces in optical tweezers setups[27,28]. The structures there are commonly based on a prescribed optical potential energy landscape in which the particles

occupy energetic minima and fluctuate according to Boltzmann statistics in equilibrium.

The assemblies created in our experiments are conceptually different as there is no external force acting on the particles[17] and more importantly, the structures do only exist in nonequilibrium. The particles need to be propelled continuously to form the assemblies much like living systems need nonequilibrium to maintain their shape and function. The fluctuations of the structures are nonequilibrium even for a vanishing feedback delay and thus they are not tied to a well-defined temperature or Boltzmann statistics. Moreover, as demonstrated for the oscillatory motion, non-zero delay in our setup leads to additional oscillations/fluctuations determined by the propulsion speed and feedback delay.

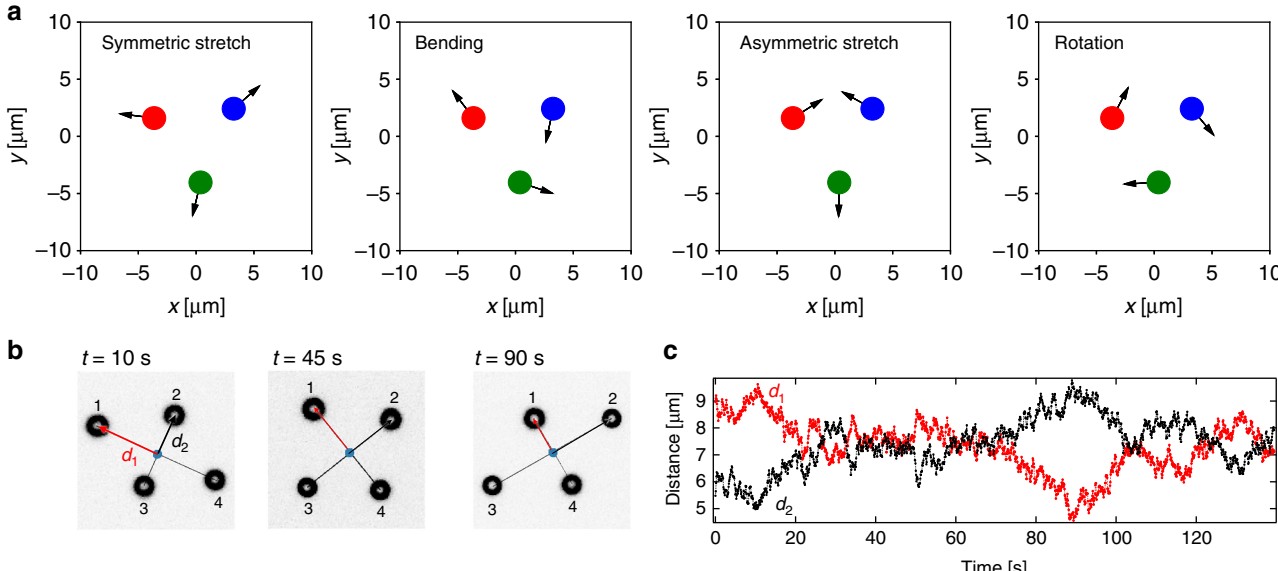

**Fig. 5** Principle components of the trimer and isomerization of the tetramer. **a** The principle components of the trimer as calculated from a principle component analysis (PCA) of the particle displacements in the center of mass coordinate system. Depicted are the symmetric stretch, bending motion, asymmetric stretch, and the rotation sorted by the magnitude of the eigenvalue found in the PCA. **b** Isomerization of the tetramer. The images represent the observed dark-field images of the 4 active particles at three different times ($t = 10$ s, $t = 45$ s, $t = 90$ s). Particles have been labeled to keep track of their position. **c** Timetrace of the distance $d_1$ of particle 1 and $d_2$ of particle 2 from the center of mass indicating the isomerization. The distances $d_1$ and $d_2$ exchange in size after a time period of about 75 s. The isomerization is driven by Brownian motion

In artificial active particle systems several sources of structure formation and local density enhancement may exist. Active particles are observed, for example, to cluster based on chemical, composition, or temperature gradients mimicking tactic behavior[34]. In the case of chemotaxis of artificial microswimmers the propulsion speed is modulated as direct physical consequence of the local concentration of a substance. In fact, a modulation of the propulsion speed will lead to local enhancement or depletion of active particle concentrations[35] in steady states as it is observed also in motility induced phase transitions with a mutual locking of particles[3,36].

With the feedback rule applied in our experiments, we now remove also the modulation of the speed as particles are constantly propelled with the same speed. Yet there is a structure forming contribution resulting from the feedback loop. It stems from the fact that we determine the particle positions in a measurement and decide on the required action, i.e., the velocity direction. It is this particular processing of the information of the particle positions which extracts entropy from the system to form the structures. Analyzing the entropy production in the system, we can identify several contributions (see Supplementary Fig. 6). On one side, there is the entropy production rate maintaining the temperature gradients. This "housekeeping" entropy production amounts to the fraction of the incident laser power absorbed by the gold nanoparticles $P_a$ divided by the temperature of the surrounding liquid $T$ to which this power is dissipated. It is required for the mobility although only a fraction of it is used during propulsion[37]. This entropy flux is still not sufficient for the structure formation. The propulsion speed and the dissipated power are constant over the trajectory of the particle no matter if it is driven on a random path or is bound in the active particle molecule. Consequently, it is not a spatial modulation of this dissipation which is causing the structure formation.

The feedback process is extracting entropy from the system by steering the propulsion direction and not modulating their speed. A theoretical analysis of the entropy fluxes can be obtained by relating the feedback process to the thermodynamics of resetting

processes as described by Fuchs et al.[38] (see Supplementary Note 4). According to that, it is the continuous loss of structure due to Brownian motion which one has to correct for with the feedback. To form a stable stationary structure, the entropy extracted per time unit in the feedback loop has to compensate at least the increase of entropy per time unit due to Brownian motion. The structure formation is thus the result of the information flow in the feedback loop only. Overall, the structure forming entropy production rate is very small as compared to all other entropy fluxes (see Supplementary Note 4) but sufficient to create well-defined active particle assemblies.

Returning to the previous example of chemotaxis, one may argue that the response of an artificial active particle to a chemical gradient may be seen as an information based interaction as well even if it is the direct physical consequence of osmotic pressure differences. The local chemical gradient is the information that is available to the particle and causes a response in form of a modulated propulsion speed. While this interpretation of physical interactions may also be valid, we restrict our definition of information based interactions to situations, where the physical interactions are irrelevant. In the well-known thought experiment of Maxwell, for example, a small daemon uses the information on the speed of particles to sort them into two boxes (slow and fast) just by actuating a shutter between the boxes. The physical interactions of the particles are irrelevant there for the appearing temperature difference between the two boxes. A correct physical description not violating the laws of thermodynamics, however, requires to include the entropy of information the daemon uses. In a very similar way, it is the information flow in the feedback loop that is required for a correct physical description of the assembled active particle molecules in our experiment.

In conclusion, propulsion speed, Brownian motion as well as the feedback related information flow shape the morphology and dynamics of these artificial self-organized active structures. They require nonequilibrium conditions to exist and contrary to structures assembled by optical tweezers they are not bound to Boltzmann statistics, equipartition or global detailed balance.

Nevertheless, the dynamics of the structures carries a number of features, which are found in equilibrium as well. Using the described method, almost any type of interaction can be designed to create large scale interacting assemblies or new self-organized shapes which may not be accessible by conventional interactions, i.e., which do not need to obey the action–reaction principle. Fundamental interaction and signaling rules for emergent complex behavior of cells up to animals may be explored with our model system in the same way as concepts of nonequilibrium thermodynamics. The applied technique provides a direct interface to machine learning algorithms including predictive information or reinforcement learning. The details of information flows in large ensembles may be studied easily and can be connected to different timescales of delayed information processing. Especially the latter type of application including coupled active feedback networks with different inherent timescales shall ignite a vast variety of research on emergent collective and crowd dynamics.

## Methods

**Sample preparation.** Samples consist of commercially available gold nanoparticle coated melamine resin particles of a diameter of 2.13 µm (microParticles GmbH, Berlin, Germany). The gold nanoparticles are covering about 30% of the surface and are between 8 and 30 nm in diameter (Supplementary Fig. 1). Glass cover slips have been dipped into a 5% Pluronic F127 solution, rinsed with deionized water and dried with nitrogen. The Pluronic F127 coating prevents sticking of the particles to the glass cover slides. Two microliters of particle suspension are placed on the glass cover slides to spread about an area of 1 cm × 1 cm to form a 3 µm thin water film. The edges of the sample have been sealed with silicone oil to prevent water evaporation.

**Microscopy setup.** Samples have been investigated in a custom-built inverted microscopy setup (Supplementary Note 5). The setup is based on an Olympus IX 71 microscopy stand. Optical heating of the active particles is carried out by a CW 532 nm laser. The laser intensity is controlled by a Conoptics 350–50 electro-optical modulator. An acousto-optic deflector (AOD) together with a 4-f system (two $f = 20$ cm lenses) is used to steer the 532 nm wavelength laser focus in the sample plane. The AOD is controlled by an FPGA (National Instruments) via a LabVIEW program. The calibration of the AOD for precise laser positioning is carried out using a 2D projection method. A Leica 100x, infinity-corrected, NA 1.4–0.7 (set to 0.7), HCX PL APO objective lens is used for focusing the 532 nm laser to the sample plane as well as for imaging the active particles. Active particles are imaged under dark-field illumination using an oil immersion dark-field condenser. The scattered light from the sample is collected with the Leica objective lens and imaged with a $f = 30$ cm tube lens to an emCCD camera (Cascade 650). A region of interest (ROI) of 200 × 200 pixels is utilized for the real time imaging, analysis and recording of the particles, with an exposure time $\Delta t_{exp}$ of 0.08 s or 0.110 s.

**Single particle tracking and real-time feedback loop.** The active particles appear as rings in dark-field microscopy images and are tracked in real time in a LabVIEW program. A Matlab node in the LabVIEW determines the centers of the particles using a Hough transform function of Matlab. The particle coordinates are used to calculate the position of the laser focus for each individual particle. During one camera exposure of $\Delta t_{exp}$, the laser is shared among the particles selected for feedback control. The switching is done using the AOD as mentioned above.

For the control experiments (Fig. 1), the controlled particle is driven back-and-forth between two prescribed target positions. Upon reaching a target, it is actively positioned there for 100 frames first, before being driven to the the target. This procedure is repeated several times (5–7) with different laser powers up to 1 mW. The particle positions around the targets are used to determine the localization error $\sigma = \sqrt{\langle \delta \mathbf{r}^2 \rangle}$, where $\delta \mathbf{r} = \mathbf{r} - \mathbf{r}_t$ is the 2-d position vector from the target to the particle. Velocities are determined by projecting the particle displacement between two subsequent frames onto the unit vector given by the laser position and the particle center $\langle v \rangle = \langle (\mathbf{r}(t + \Delta t_{exp}) - \mathbf{r}(t)) \cdot \mathbf{e}_{lp} \rangle / \Delta t_{exp}$, where $\mathbf{e}_{lp} = (\mathbf{r} - \mathbf{r}_{laser})/|\mathbf{r} - \mathbf{r}_{laser}|$.

**Principle component analysis of the dynamics.** The experiments sample the position vector $\mathbf{r}_i(t)$ for each of the $N$ particles in $N_{steps} = t_{meas}/\Delta t_{exp} + 1$ time instants. The particle coordinates are converted into the center of mass frame of the structure. In two dimensions, we obtain $N_{data} = 2 \times N \times N_{steps}$ data points, which we analyze using the principal component analysis.

First, we construct the $2N$ time-series of displacements of the individual degrees of freedom in our experiment (coordinates of the individual $N$ particles). Second, we put all these displacements for a given $t$ into a single vector: $\mathbf{S}(t) = (\Delta x_1(t), \Delta y_1(t),..., \Delta x_N(t), \Delta y_N(t))$. Then, we construct the matrix $\mathbf{X}$ containing in its lines the vectors $\mathbf{S}(t)$ corresponding to the individual measurement times, so the element

$[i, j]$ of this matrix is given by $X_{ij} = S_j((i-1)\Delta t_{exp})$. The matrix $\mathbf{X}$ thus has $2N$ columns and $N_{steps} - 1$ rows. The matrix $\mathbf{X}$ is used to calculate the covariance matrix $\mathbf{M} = \mathbf{X}^T \mathbf{X}$. This matrix $\mathbf{M}$ is a symmetric $2N \times 2N$ matrix with the elements $M_{ij} = \sum_{k=1}^{N_{steps}-1} X_{ki} X_{kj} = \sum_{k=1}^{N_{steps}-1} S_i((k-1)\Delta t_{exp}) S_j((k-1)\Delta t_{exp})$. The diagonal thus contains the variances corresponding to the degrees of freedom measured in the experiment, i.e., of the displacements of the positions of the individual particles.

We determine all $2N - 2$ nonzero eigenvalues $A_i$ and normalized eigenvectors $\mathbf{V}_i$ of the matrix $\mathbf{M}$.

The individual eigenvectors determine new $2N - 2$ collective degrees of freedom, which are mutually independent. For example for the dimer, the eigenvector with the largest eigenvalue determines the vibrational mode and the corresponding collective coordinate is proportional to the vector connecting the two particles.

The vector form of the time series corresponding to the mode given by the column eigenvector $\mathbf{V}_i$ can be obtained by projecting the eigenvector $\mathbf{V}_i$ onto the matrix $\mathbf{X}$, i.e., $\mathbf{K}_i = \mathbf{X} \cdot \mathbf{V}_i$. The elements of the $\mathbf{K}_i$ represent the time series $A_i(t)$ of the motion along the eigenvector $\mathbf{V}_i$. To access the dynamics of the mode we calculate its autocorrelation $C_i(t) = \langle A_i(\tau) A_i(\tau + t) \rangle_\tau / \langle A_i(\tau)^2 \rangle_\tau$. The subscript $\tau$ denotes that the correlation function is obtained from the time average (Supplementary Note 2).

## Data availability

All data is available from the corresponding author on request.

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

## Acknowledgements

Discussions with J. Shaevitz (Princeton University), K. Kroy (Universität Leipzig) and help with the sample preparations by D. Cichos (Berlin) are acknowledged. H.Y. and U. K. acknowledge support by the Betty and Gordon Moore foundation (grant # 4741). V.H. is supported by a Humboldt grant of the Alexander von Humboldt Foundation and by the Czech Science Foundation (project No. 17-06716S). F.C. is supported by grant CI 33/ 16-1 and the CRC TRR 102 "Polymers under multiple constraints" of the German Research Foundation (DFG).

## Author contributions

U.K. and F.C. designed the experiments. U.K. and F.C. performed the experiments and analyzed the data. V.H. contributed to the theoretical analysis. H.Y. provided the experimental equipment. U.K., V.H., H.Y., and F.C. wrote the manuscript. All authors discussed the results and commented on the manuscript.

## Additional information

**Competing interests:** The authors declare no competing interests.

