## [Peer Review file · Nature Communications]

Reviewers' comments:

Reviewer #1 (Remarks to the Author):

The authors present a novel experimental system, consisting of colloidal particles that achieve thermophoresis in response to an externally imposed laser signal. The laser signal is coupled to the microscope system through a feedback loop and can be programmed to create stimuli in response to individual and/or collective dynamics of the colloids, which then in turn leads to more complex collective behaviors.

I think this is an interesting experimental study, which promises new opportunities for interfacing computational methods with active matter. The approach is similar in spirit to recent work on coupled chemical oscillators (Nature Physics 14: 282–285, 2018), though very different in terms of technical implementation. The manuscript is well written and should be accessible to a broad readership. I therefore believe this paper could be suitable for Nature Communications.

A few minor comments:

1) What is the limit in terms of achievable system sizes / particle numbers?

In my opinion, it would be interesting if the authors could discuss this point in more detail and, if experimentally possible, present a demonstration for larger particle numbers.

2) The authors mention machine learning several times in abstract and text, but do not actually demonstrate an ML application. I would perhaps restrict the ML reference to the conclusions section.

Reviewer #2 (Remarks to the Author):

This manuscript reports an experimental investigation of the self-organization of active particles. The authors assert that they are, for the first time, exploring self-organization of active particles based on “information flow” between particles and not direct physical interaction between the particles. The latter approach is what the authors indicate has been the focus of past investigations. The experimental approach involves the local heating of particles (decorated with gold nanoparticles to promote absorption of light and local heating) and consequent thermophoresis. The physical mechanism of driving the particles into motion is not new, but what the authors claim is novel is their feedback control of local heating in ensembles of particles based on a simple algorithm that seeks to maintain interparticle separations at a set point value.

Overall, the experimental system with feedback is elegantly simple, and it does seem quite versatile. It is also interesting to see what types of collective behaviors emerge with their simple feedback control. However, what is observed is fairly intuitive (not unexpected), and the level of analysis in terms of significant new insights regarding information flow and entropy is somewhat limited and disappointing given the wide ranging statements of significance made in the Introduction. The PCA analysis at the end of the paper (p 9) that identifies modes of vibration/rotation is also fairly elementary. It is not clear that significant new insights emerge.

I also think that there are prior examples where information flows between active particles do influence their collective behavior. For example, many chemical systems (dissolving oil droplets of Bahr, or H₂O₂-driven systems of Sen/Whitesides) generate chemical gradients that lead to collective behaviors. These can be equally viewed as systems in which information flows between particles in the

absence of direction interparticle interactions (e.g., Coulomb interactions). I agree that the precedents don't have the same versatility of the system reported in this paper, but they are precedent for systems in which organization emerges from "information flow".

In addition, there is a literature in which optical traps have been manipulated based on feedback control (e.g., interparticle separation) to create organized arrays. The systems are not "active" according to the definitions used in this paper, but they are colloidal, and organization emerges from information flow between the colloids much in the way that it does in this paper. As example of this type of work would be Michael Bevan from Johns Hopkins University.

While I have some reservations about the significance of the results reported in this paper in terms of developing new principles or insights into self-organization in active systems, the experimental system is an interesting one and future studies with it may lead to important conceptual advances. I don't see those conceptual advances, however, being present in this paper. The work is carefully executed with statistical analyses supporting conclusions.

Reviewer #3 (Remarks to the Author):

In their manuscript, Khadka et al. study experimentally an active particle system where feedback interaction rules among particles are taken into account.

In the considered experimental setup, a melamine resin sphere with 30% of the surface uniformly covered by gold nanoparticles becomes self-propelled when illuminated with a laser beam. The laser beam generates a temperature gradient on the surface of the sphere inducing self-thermophoretic motion. The self-propulsion velocity can be tuned at single particle level by tracking the particle center. Since the feedback interaction rule can be designed arbitrary for generating non-mechanical interactions among particles, it can play the role of a sensing interaction.

In particular, they consider non-mechanical interactions that couple particle positions and self-propulsion velocity.

In this way, opportune interaction rules can be employed for tuning collective behaviors. For instance, one can generate static patterns, e. g., arranging particles in static structures, or dynamic patterns, e. g., moving the structures towards a target area.

I find the results presented in the paper really interesting and the manuscript is well written. Moreover, the topic is of interest to a wide audience of scientists working not only in Active Matter but also in out-of-equilibrium and biological physics.

However, there are some crucial points that should be carefully addressed by the authors. I would recommend its publication in Nature Communication after revision. My main concerns are given below.

Comments

1. Even though the physics underlining the pattern formation explored here is different from the physics in holographic optical tweezers, multiple particle trapping by speckle patterns, and other trapping techniques, the emerging phenomenology is quite similar.

The authors write about that at p. 6, "As compared to optical tweezers, the applied scheme does not

involve external forces, but just dissipative fluxes and is thus physically different from the common trapping”.

I understand that the two situations are physically different, however, in both cases, manipulation happens through external fields. Moreover, in both cases, a continuous feedback and computer control are necessary [1,2]. In particular, multiple trapping and tracking techniques could be employed for driving particles and assembling structures [2]. It is reasonable to think that even in that case of holographic trapping, one can define an information flow needed to bound particles into ordered structures.

What are the advantages of considering self-propelled objects instead optically trapped passive particles? The new physics is just related to the absence of external forces or there is something more?

2. At p. 4, the authors write “Different from standard active particles, the timescale of the rotational diffusion of the particle is irrelevant due to the missing particle asymmetry.”

If I understand correctly, this means that the self-propelled particles considered here are always superdiffusive on the experimental time scale. Could the author include a plot of the mean-square displacement showing that rotational diffusion is negligible?

3. I would like to suggest to the authors to improve the discussion about entropy production at p. 7.

I agree that “there is a continuous entropy production in the system which comprises different contributions. The first contribution is a constant entropy production rate maintaining the temperature gradients and propelling the particles, which is required for the mobility but not sufficient for the structure formation.”

This contribute is due to the incident laser beam that generates the temperature gradient on the particle surface. Am I correct?

Since in SI, p. 3, they estimate the tangential temperature gradient, could they also estimate that entropy production term?

Then, they write “The emerging artificial molecules are the result of the position measurement and feedback extracting entropy from the system by steering the propulsion direction.” In this process seems to contribute also information exchange between particles, is there also any entropy production related to information entropy?

Finally, “To form a stable stationary structure, the entropy extracted per time unit in the feedback loop has to compensate at least the increase of entropy per time unit due to Brownian motion. The structure formation is thus the result of the information flow in the feedback loop only.” According to that picture, the feedback loop makes the work necessary to stabilize the structure. Is that interpretation correct?

4. Some results shown in Fig. 3, panel B seems to be obtained through numerical simulations. However, I did not find any discussion about numerical simulations in the manuscript. How have these data been obtained? Could the author add a section in methods or SI describing how the simulations are performed and compared with experiments?

Minor Points

1. The meaning of the symbols appearing in Fig. 1 (A), i. e., ω_0 , Δx should be introduced in the main text and/or in the caption.
2. I would suggest to include a bar into the figures indicating 1 μm length in Fig. 1 (A), Fig. 2 (A), Fig. 3 (A), and Fig. 1 (A) in SI.
3. I would suggest to the authors to label each panel/image composing a figure, e. g., Fig 1., A, B, C, D.
4. p. 4 "left graph of Figure 1D" there is no panel D in Fig. 1.
5. P. 6, at the very beginning "as shown in Figure." What figure?
6. P. 7, maybe a reference to panels in "Figure 3 highlights snapshots of..." could be useful to the reader.
7. P. 7, I would suggest adding a citation about PCA.
8. P. 8 "...is a fundamental feature of a time-delayed negative feedback" needs a citation.
9. SI, Fig. 2. The symbols in the figure are not introduced/defined into the caption.
10. SI, p. 4. "Accordingly, to obtain the maximum velocity the laser has to be placed at about a distance corresponding to the particle radius." Did they check in the experiment?
11. SI, Fig. 3. I suggest introducing labels for the panels.
12. SI, Fig. 5. AOM -> AOD. The abbreviation FPGA does not appear in the figure.
13. SI, p. 9. Ring radius is not specified in the caption of movie 1 and 2, i. e., "radius XX..."
14. In Movie 1, particles overlap each other. Could the author comment about that?

[1] Jennifer E. Curtis, Brian A. Koss, David G. Grier, "Dynamic holographic optical tweezers", *Optics Communications* 207, 169-175 (2002).

[2] Miles Padgett, and Roberto Di Leonardo, "Holographic optical tweezers and their relevance to lab on chip devices", *Lab. Chip.* 11, 1196-1205 (2011).

Reviewer #1:

The authors present a novel experimental system, consisting of colloidal particles that achieve thermophoresis in response to an externally imposed laser signal. The laser signal is coupled to the microscope system through a feedback loop and can be programmed to create stimuli in response to individual and/or collective dynamics of the colloids, which then in turn leads to more complex collective behaviors.

I think this is an interesting experimental study, which promises new opportunities for interfacing computational methods with active matter. The approach is similar in spirit to recent work on coupled chemical oscillators (Nature Physics 14: 282–285, 2018), though very different in terms of technical implementation. The manuscript is well written and should be accessible to a broad readership. I therefore believe this paper could be suitable for Nature Communications.

Thank you very much for your review!

A few minor comments:

1) What is the limit in terms of achievable system sizes / particle numbers?

In my opinion, it would be interesting if the authors could discuss this point in more detail and, if experimentally possible, present a demonstration for larger particle numbers.

The maximum number of particles that can be addressed depends on three things. The first is the speed of the tracking algorithm, the second is the complexity of the computation and the third is the speed of the response system i.e. the laser steering. Our current experiments involve up to 30 particles to be controlled within 50 ms without having optimized any of these procedures. We are convinced that we will be able to control at least 100 particles with a steering by a spatial light modulator and an optimized tracking. So far we have not carried out more complex calculations than an implementation of reinforcement learning, which is contained in a manuscript that is currently under review.

2) The authors mention machine learning several times in abstract and text, but do not actually demonstrate an ML application. I would perhaps restrict the ML reference to the conclusions section.

Yes, a machine learning application is not part of the current manuscript. We have recently submitted an additional manuscript on this topic. The preprint is available at: <https://arxiv.org/abs/1803.06425>

Reviewer #2:

This manuscript reports an experimental investigation of the self-organization of active particles. The authors assert that they are, for the first time, exploring self-organization of active particles based on “information flow” between particles and not direct physical interaction between the particles. The latter approach is what the authors indicate has been the focus of past investigations. The experimental approach involves the local heating of particles (decorated with gold nanoparticles to promote absorption of light and local heating) and consequent thermophoresis. The physical mechanism of driving the particles into motion is not new, but what the authors claim is novel is their feedback control of local heating in ensembles of particles based on a simple algorithm that seeks to maintain interparticle separations at a set point value.

I also think that there are prior examples where information flows between active particles do influence their collective behavior. For example, many chemical systems (dissolving oil droplets of Bahr, or H₂O₂-driven systems of Sen/Whitesides) generate chemical gradients that lead to collec-

tive behaviors. These can be equally viewed as systems in which information flows between particles in the absence of direction interparticle interactions (e.g., Coulomb interactions). I agree that the precedents don't have the same versatility of the system reported in this paper, but they are precedent for systems in which organization emerges from "information flow".

We respectfully disagree that other artificial systems, which demonstrate different types of taxis (chemical, thermal or other) do show collective behavior due to information flows as well. While this may be possible in bacterial systems, where the chemical signal is analyzed and translated into an action i.e. of cilia or flagella, active particles or droplets do respond to chemical gradients by (thermodynamic) forces i.e. due to phoretic effects for example. The presence of chemical gradients does in that sense not correspond to information. Particles or droplets in those cases drive faster or slower depending on the size chemical gradients. Therefore the dissipative fluxes that drive the particle are modulated in strength depending on the position of the particle. According to the dependence of the probability density for finding an active particle in space, this leads to local accumulation where particles are slow i.e. when they collide. Such a situation does not exist in our case. Particles are driven constantly with the same speed, they are not modulated in their speed. There is no difference in the dissipation on a random path or in the bound state as the speed is constant. The bound state must therefore be solely created from the information on the position and not by modulating the speed. This is a fundamental difference as compared to any other study. In addition, our system relies on a retarded action, which is inherent to the many feedback loops of living systems, but of no relevance for the chemical particles mentioned. To imitate a similar behavior with optical tweezers, requires a driving at constant speed there as well. However, the fundamental difference of a propulsion without an external force in our case and with external force in the tweezer case always remains. An active particle is a non-equilibrium system even if it stands still. In this case it just pumps the liquid around it. This is not the case for a trapped particle in a tweezer. We have tried to sharpen this issue in the manuscript in the marked regions and by including a section on the analysis of entropy fluxes in the supplement.

In addition, there is a literature in which optical traps have been manipulated based on feedback control (e.g., interparticle separation) to create organized arrays. The systems are not "active" according to the definitions used in this paper, but they are colloidal, and organization emerges from information flow between the colloids much in the way that it does in this paper. As example of this type of work would be Michael Bevan from Johns Hopkins University.

We thank the referee for pointing out prior studies involving manipulation of multiple colloids using optical trap arrays. There has been indeed beautiful work done with the help of optical tweezers also for larger arrays of particles and the self-organization into structures. Some of our intermediate results where active particles were structured into patterned arrangements in order to demonstrate the multi-particle experimental controllability share a similar spirit to the outcomes of the optical trap array systems described by the Referee. We have added two new references for highlighting this field. We are convinced that similar processes as presented in our manuscript could be possible with the help of optical tweezers, but with the fundamental difference that our active particles are not controlled by external forces and therefore propelled by force dipoles, while in optical tweezers the particles will always be controlled by external forces. In other words, the sample is in our case force free and it is not for an optical tweezer. To achieve at least the same processes as presented with an optical tweezer, the particles would have to be driven with a constant force. As mentioned in the previous response, active particles are non-equilibrium systems and behave differently for example if they collide with walls or other particles. If two self-propelled particles collide and therefore do not move actively, they still dissipate the energy by creating a flow field, which is not present when no force is acting on a tweezed particle.

Overall, the experimental system with feedback is elegantly simple, and it does seem quite versatile. It is also interesting to see what types of collective behaviors emerge with their simple feedback control. However, what is observed is fairly intuitive (not unexpected), and the level of analysis in terms of significant new insights regarding information flow and entropy is somewhat limited and disappointing given the wide ranging statements of significance made in the Introduction. The PCA analysis at the end of the paper (p 9) that identifies modes of vibration/rotation is also fairly elementary. It is not clear that significant new insights emerge.

While I have some reservations about the significance of the results reported in this paper in terms of developing new principles or insights into self-organization in active systems, the experimental system is an interesting one and future studies with it may lead to important conceptual advances. I don't see those conceptual advances, however, being present in this paper.

We are very grateful for these comments of the reviewer. In fact, the conceptual advances have been partly mentioned in our response to the other comments of the reviewer. Our experiments remove a modulation of the dissipation as a source of structure formation from the system by keeping the speed and thus the dissipation constant. The particle could travel with this speed any other path, which is not creating a bound state and still dissipate the same amount of energy per time during propulsion. It is, as compared to other systems like optical tweezers, only the information flow in the feedback loop, which is responsible for structure formation. The structures exist only in non-equilibrium. Information on the system position is gained by the measurement and lost during the exposure due to Brownian motion. Moreover, this information flow leads to phenomenologies of the dynamics, which are expected for force driven systems and, as the reviewer mentions, quite intuitive but their origin is in our case entirely different. We therefore suggest, that a number of structure formation processes observed in living systems may also depend purely on information flows rather than on physical interactions. We have added a more detailed analysis of the entropy fluxes to the supplementary information to support our points. Also the discussion in the main text has been modified to express these points.

Reviewer #3:

In their manuscript, Khadka et al. study experimentally an active particle system where feedback interaction rules among particles are taken into account.

In the considered experimental setup, a melamine resin sphere with 30% of the surface uniformly covered by gold nanoparticles becomes self-propelled when illuminated with a laser beam. The laser beam generates a temperature gradient on the surface of the sphere inducing self-thermophoretic motion. The self-propulsion velocity can be tuned at single particle level by tracking the particle center. Since the feedback interaction rule can be designed arbitrary for generating non-mechanical interactions among particles, it can play the role of a sensing interaction.

In particular, they consider non-mechanical interactions that couple particle positions and self-propulsion velocity.

In this way, opportune interaction rules can be employed for tuning collective behaviors. For instance, one can generate static patterns, e. g., arranging particles in static structures, or dynamic patterns, e. g., moving the structures towards a target area.

I find the results presented in the paper really interesting and the manuscript is well written. Moreover, the topic is of interest to a wide audience of scientists working not only in Active Matter but also in out-of-equilibrium and biological physics.

However, there are some crucial points that should be carefully addressed by the authors. I would recommend its publication in Nature Communication after revision. My main concerns are given below.

We thank the referee very much for the comments and suggestions. Please find a detailed response below.

Comments

1. Even though the physics underlining the pattern formation explored here is different from the physics in holographic optical tweezers, multiple particle trapping by speckle patterns, and other trapping techniques, the emerging phenomenology is quite similar.

The authors write about that at p. 6, “As compared to optical tweezers, the applied scheme does not involve external forces, but just dissipative fluxes and is thus physically different from the common trapping”.

I understand that the two situations are physically different, however, in both cases, manipulation happens through external fields. Moreover, in both cases, a continuous feedback and computer control are necessary [1,2]. In particular, multiple trapping and tracking techniques could be employed for driving particles and assembling structures [2]. It is reasonable to think that even in that case of holographic trapping, one can define an information flow needed to bound particles into ordered structures.

What are the advantages of considering self-propelled objects instead optically trapped passive particles? The new physics is just related to the absence of external forces or there is something more?

We thank the Referee for raising this issue. There has been a lot of excellent work done with the help of different types of optical tweezers and we have added more references to acknowledge this work. As mentioned by the Referee, the first difference is of course the physical propulsion mechanism, which keeps our system force free even though the laser is used as energy source. The latter is rather due to a current lack of technology putting the energy source inside the swimmer itself and keeping the control.

If we ignore these physical differences we think that a similar situation may be constructed with optical tweezers itself. In this case the particles have to be actuated in a constant force mode as our particles are driven with a constant speed to remove a modulation of dissipation due to different propulsion speeds. A main difference of the non-equilibrium and the externally driven systems is in addition, that active particles still dissipate when they do not move, while externally driven system will not dissipate if no force is acting.

2. At p. 4, the authors write “Different from standard active particles, the timescale of the rotational diffusion of the particle is irrelevant due to the missing particle asymmetry.”

If I understand correctly, this means that the self-propelled particles considered here are always superdiffusive on the experimental time scale. Could the author include a plot of the mean-square displacement showing that rotational diffusion is negligible?

Thank you very much for this remark. The gold nanoparticles cover 30 % of the particle surface. As a result of particle preparation, there might be of course inhomogeneities in the local density of nanoparticles at the particle surface. In this case the propulsion velocity might be modulated with the orientation. However, in our experiments we have not observed a visible influence of the orientation. We have added a short section displaying the mean squared displacement for three different driving periods of the experiments shown in the Supplementary Figure 5b to the supplement. The results do well agree with the expected parabolic behavior of the mean squared displacement.

3. I would like to suggest to the authors to improve the discussion about entropy production at p. 7.

I agree that “there is a continuous entropy production in the system which comprises different contributions. The first contribution is a constant entropy production rate maintaining the temperature gradients and propelling the particles, which is required for the mobility but not sufficient for the structure formation.”

This contribute is due to the incident laser beam that generates the temperature gradient on the particle surface. Am I correct?

Yes, both the normal and tangential temperature gradients are contributions to this entropy production.

Since in SI, p. 3, they estimate the tangential temperature gradient, could they also estimate that entropy production term?

Yes, the constant entropy production rate maintaining the temperature gradient is generated due to the laser beam. Since the system operates in a non-equilibrium steady-state, its energy is constant and all the power P_a transferred by the laser to the system (colloids + water) is eventually dissipated. The corresponding entropy production can be estimated as $\dot{S}_B = P_a/T$. Here, T is the temperature of the reservoir (for example walls of the experimental setup), where all the energy influx to the system is eventually transferred and dissipated. P_a is the fraction of the total incident laser power P_{heat} that is absorbed by the gold nanoparticles. It can be determined by the absorption cross section of the gold nanoparticles and the number of nanoparticles illuminated by the laser. If we assume the 30% coverage and an average diameter of the gold nanoparticles of 10 nm (as specified by the manufacturer), Mie theory gives an approximate absorption cross section of 10 nm^2 and we find for an incident power of 1 mW (focused to a beam waist of 250 nm) an absorbed power of about $9 \mu\text{W}$. This gives an entropy production of 30 nJ/K/s . We have also carried measurements of the absorbed power by placing the particles in a nematic liquid crystal (5CB). Heating the gold particles on the surface melts the nematic phase and an isotropic bubble is formed around the particle. From the size of the bubble we have estimated an absorbed power of $80 \mu\text{W}$. According to that the average gold nanoparticle diameter should be rather close to 18 nm. In that case, the entropy production rate shall amount for 270 nJ/K/s . For more exact data, we still need more detailed electron microscopy measurements of the gold nanoparticle size. From this absorbed power, most of the power is required for „housekeeping“, maintaining the temperature gradient. Only a fraction of this power is finally used to propel the particle. Similar as for Janus particles, the efficiency is very low. An analysis for Janus particles with the same propulsion mechanism is contained in [1]. If we estimate the power required for a particle to propel by $F=6\pi\eta Rv^2$, we find $2e-20 \text{ W}$. The propulsion efficiency is therefore on the order of $2.5e-16$. This in principle also allows the calculation of the information entropy production rate.

[1] Bregulla, A. P. & Cichos, F. Size dependent efficiency of photophoretic swimmers. *Faraday Discuss.* 184, 381–391 (2015).

A more detailed description of all the energy fluxes and the motion of the symmetric swimmer is currently summarized in a separate manuscript. For the current manuscript, we have tried to improve the entropy discussion in the main text and added a section with a formal analysis of the entropy production to the supplement. In fact, the extracted entropy per time unit has to correspond for the dimer particle to twice the power used to propel the particle divided by the temperature of the bath.

Then, they write “The emerging artificial molecules are the result of the position measurement and feedback extracting entropy from the system by steering the propulsion direction.” In this process seems to contribute also information exchange between particles, is there also any entropy production related to information entropy?

Yes. We have added a more detailed analysis of the entropy production rates to the supplementary information. As it turns out there, the entropy extraction for a dimer corresponds to twice the entropy production for propelling a single particle. This entropy production rate is required to correct for the displacement of the particles due to diffusion. The factor of two corresponds to the number of particles.

Finally, “To form a stable stationary structure, the entropy extracted per time unit in the feedback loop has to compensate at least the increase of entropy per time unit due to Brownian motion. The structure formation is thus the result of the information flow in the feedback loop only.” According to that picture, the feedback loop makes the work necessary to stabilize the structure. Is that interpretation correct?

Yes, in a sense. The necessary physical work keeping the structure is done by the laser as it supplies the energy. But this would not be sufficient to form the structure alone, without the information about desirable directions in which this energy should be released. There is also some work connected with processing this information. While the lower bound for that is imposed by the Landauer principle, in our case it can be simply determined by the power input of the feedback device (computer etc.) and thus it is much larger than the lower bound given by the Landauer principle.

Overall we have modified the entropy section in the main text. The major new information on the entropy has been added to the supplementary information.

4. Some results shown in Fig. 3, panel B seems to be obtained through numerical simulations. However, I did not find any discussion about numerical simulations in the manuscript. How have these data been obtained? Could the author add a section in methods or SI describing how the simulations are performed and compared with experiments?

We have added a section describing the simulations in the supplementary information.

Minor Points

1. The meaning of the symbols appearing in Fig. 1 (A), i. e., ω_0 , Δx should be introduced in the main text and/or in the caption. *done*
2. I would suggest to include a bar into the figures indicating 1 μm length in Fig. 1 (A), Fig. 2 (A), Fig. 3 (A), and Fig. 1 (A) in SI. *done*
3. I would suggest to the authors to label each panel/image composing a figure, e. g., Fig 1., A, B, C, D. *done*
4. p. 4 “left graph of Figure 1D” there is no panel D in Fig. 1. *done*
5. P. 6, at the very beginning “as shown in Figure.” What figure? *corrected*
6. P. 7, maybe a reference to panels in “Figure 3 highlights snapshots of...” could be useful to the reader. *done*
7. P. 7, I would suggest adding a citation about PCA. *done*
8. P. 8 “...is a fundamental feature of a time-delayed negative feedback” needs a citation. *done*
9. SI, Fig. 2. The symbols in the figure are not introduced/defined into the caption. *done*
10. SI, p. 4. “Accordingly, to obtain the maximum velocity the laser has to be placed at about a distance corresponding to the particle radius.” Did they check in the experiment? *Yes, that will be part of a forthcoming paper.*
11. SI, Fig. 3. I suggest introducing labels for the panels. *done*
12. SI, Fig. 5. AOM \rightarrow AOD. The abbreviation FPGA does not appear in the figure. *done*
13. SI, p. 9. Ring radius is not specified in the caption of movie 1 and 2, i. e., “radius XX...” *done*
14. In Movie 1, particles overlap each other. Could the author comment about that?

Particles overlap due to the sample thickness being larger than twice the particle diameter in that experiments. The employed Hough transform in the particle tracking is robust against some of the overlapping of particles. In all other cases the laser is positioned at places where a particle has been localized. In cases where two particles overlap in the image and the tracker is finding only one particle, both particles are heated with the same laser spot and active and Brownian motion is separating the particles.

Thank you very much for the very detailed list!! We appreciate this very much! We hope to have addressed all issues.

[1] Jennifer E. Curtis, Brian A. Koss, David G. Grier, “Dynamic holographic optical tweezers”, *Optics Communications* 207, 169-175 (2002).

[2] Miles Padgett, and Roberto Di Leonardo, “Holographic optical tweezers and their relevance to lab on chip devices”, *Lab. Chip.* 11, 1196-1205 (2011).

Reviewers' comments:

Reviewer #1 (Remarks to the Author):

The authors have satisfactorily addressed my comments in the revised manuscript (as well as those by the other reviewers, as far as I can see). As already expressed in my first report, this is an interesting new type of experiment that has the potential of opening a new area in the field of active matter research. I am therefore happy to recommend this revised version for publication.

Reviewer #2 (Remarks to the Author):

I have read the revised manuscript and the response of the authors to the reviewer comments. The revisions to the manuscript are helpful, but I struggle to understand some key aspects of the authors' response:

First, it seems that the authors are taking a very restrictive (contrived?) definition of information flow in order to differentiate their work from past studies with optical traps and active particles. There are many examples of active particles that do not directly interact yet exhibit collective behaviors - they organize as a result of information flow between the particles that is mediated by overlapping chemical gradients ("wakes"). The authors argue in their letter of response that chemical gradients in this case do not correspond to information - but I argue that the chemical gradients reflect the positions (information) about the other particles in the system just as the thermal gradients generated in the authors' experimental system reflect the positions of other particles in their systems. I do not see the fundamental difference. Additionally, as pointed out by the authors, the dissipative fluxes that drive the particles are modulated in strength depending on the position of the particles in the chemical system - I do not see why this fact prevents one from concluding that the organization results from information flow. Similarly, I do not understand why driving the particles at constant speed is necessary to conclude that the organization results from information flow.

Second, in the experiments reported in this paper, local heating of the particle surfaces generates a force that acts on the particles. I do not understand the authors' statement that their system differs from optical traps in that "there is no external force acting on the particles" (page 9 of revised manuscript). It seems to me that both optical traps and the experiments reported in this paper generate external forces that act on particles to drive their organization (the origins of the forces are different).

These points need to be clarified to establish the fundamental significance of the results reported in this paper.

Reviewer #3 (Remarks to the Author):

The authors carefully addressed my concern. I recommend publication of the manuscript in Nature Communication.

Reviewer #2 (Remarks to the Author):

I have read the revised manuscript and the response of the authors to the reviewer comments. The revisions to the manuscript are helpful, but I struggle to understand some key aspects of the authors' response:

First, it seems that the authors are taking a very restrictive (contrived?) definition of information flow in order to differentiate their work from past studies with optical traps and active particles. There are many examples of active particles that do not directly interact yet exhibit collective behaviors - they organize as a result of information flow between the particles that is mediated by overlapping chemical gradients ("wakes"). The authors argue in their letter of response that chemical gradients in this case do not correspond to information – but I argue that the chemical gradients reflect the positions (information) about the other particles in the system just as the thermal gradients generated in the authors' experimental system reflect the positions of other particles in their systems. I do not see the fundamental difference.

Additionally, as pointed out by the authors, the dissipative fluxes that drive the particles are modulated in strength depending on the position of the particles in the chemical system – I do not see why this fact prevents one from concluding that the organization results from information flow. Similarly, I do not understand why driving the particles at constant speed is necessary to conclude that the organization results from information flow.

Thank you very much for the very interesting discussion of this issue. We think that information is always linked to a physical representation as for example discussed by Landauer [1]. In that sense, the presence of a chemical gradient or the presence of very dilute molecules represents an information that is available. We agree that a system may respond to this information to show a chemotactic behavior. As mentioned in our previous reply, we would argue that bacteria for example may respond to the information provided by a chemical gradient by processing this information and taking actions corresponding to it (see [2] for example). In this sense we also agree that chemotaxis can be related to information flows. Still, not all systems, which show a motion along a chemical, temperature or concentration gradient are moving because of information flows. The classic thought experiment of the Maxwell daemon is emphasizing the meaning of information. A Maxwell daemon is using information on the speed of particles helping to sort them into fast and slow particles by using a shutter. There the information gained by the daemon is required and has to be included in a correct physical description of the system, since there is no physical force acting.

In the same way, we argue for chemotaxis. There may be systems using the information of gradients for chemotaxis, but in the literature of colloidal particles (also artificial active particles), the motion in the chemical gradient is, to our best knowledge, due to interfacial forces, osmotic pressure gradients or chemical reactions which physically bias the motion. The colloids do not "decide" what to do based on the information contained in chemical gradients, they are directly driven by the physical forces created by the gradients and thus it is sufficient to know these forces to predict the motion.

In the presented experiments, we know the physical forces in the system and they are constant at all relevant positions of the particles, yet a structure is formed. We propel the particles at a constant speed without a spatial modulation of the temperature and thus the structures appear in our experiments solely due to the changes of particle self-propelling directions based on the information about positions of other particles. In a sense, our particles decide what to do based on this information (in fact a computer decides for them).

To explain the structure seen in the experiments it is not sufficient to know only the physical forces in the sample but one has to include the information flow in the feedback loop.

We consider the constant propulsion speed without a spatial modulation of the temperature in order to be sure that no structure will arise if the feedback is off. This is, as we think, the best way to prove that only information from the feedback loop is relevant not additional thermophoretic interactions between the particles. Overall, we, therefore, agree with the referee, that under certain conditions chemotaxis may involve an information flow, but the current examples of collective motion of colloidal particles do not belong to this class.

Again, we are very grateful to the reviewer for discussing this issue.

[1] Landauer, R. The physical nature of information. *Physics Letters A* 217, 188–193 (1996).

[2] Micali, G. & Endres, R. G. Bacterial chemotaxis: information processing, thermodynamics, and behavior. *Curr. Opin. Microbiol.* 30, 8–15 (2016).

Second, in the experiments reported in this paper, local heating of the particle surfaces generates a force that acts on the particles. I do not understand the authors' statement that their system differs from optical traps in that "there is no external force acting on the particles" (page 9 of revised manuscript). It seems to me that both optical traps and the experiments reported in this paper generate external forces that act on particles to drive their organization (the origins of the forces are different).

The fundamental physical difference between a swimmer and an object moved by external forces is in their flow field. The swimmer is pushed forward by pushing the liquid backwards. Therefore a force balance arises between the liquid and the particle. The flow field generated by this process is essentially coming from a force dipole and it has no long range components (only $1/r^2$ or $1/r^3$). For a swimmer (or a boat on a lake with a motor) no external force is required to move and the system (swimmer + liquid) is force free. In the case of a colloid driven by an optical tweezer, the force driving the particle is coming from outside the sample. The laser field is the rope dragging the particle through the liquid. This external force is then balanced by the Stokes friction in the liquid when moving the particle at constant speed. The flow field generated in this case is different (long $1/r$) as it corresponds to the action of a point force.

In the case of the thermophoretic swimmer, the liquid motion comes from a thermo-osmotic boundary flow at the surface of the swimmer [3]. This occurs due to the temperature gradient along the particle surface. This thermo-osmotic flow is the reason for the self-thermophoretic motion of the particle (see [4] for example). The laser is therefore only providing the energy to heat the surface and not an external force. This is meanwhile well established in the literature [4] and we have only added some details about the particular type of swimmer, which is contained in the supplement and manuscript.

To better guide the reader to this literature, we have added the two references below to the main text.

[3] Bregulla, A. P., Würger, A., Günther, K., Mertig, M. & Cichos, F. Thermo-Osmotic Flow in Thin Films. *Phys Rev Lett* 116, 188303 (2016).

[4] Kroy, K., Chakraborty, D. & Cichos, F. Hot microswimmers. *Eur Phys J: Special Topics* 225, 2207–2225 (2016).

REVIEWERS' COMMENTS:

Reviewer #2 (Remarks to the Author):

I have read the authors' thoughtful response to my comments. Overall, however, I am yet to be convinced by their discussion claiming a fundamental distinction between colloids undergoing directed motions in a chemical gradient and the computer-driven motion of their particles based the locations of other particles (I do see a difference in terms of versatility). Specifically, a colloid can be engineered to respond to a chemical gradient in a manner that is dependent on the specified structure of the colloid. I don't see this type of engineered response being distinct from the engineered computer-driven response based on the positions of other particles. In the latter case, the computer has a structure that dictates the outcome, just as the colloid has a structure that dictates the outcome. The structure of both can be varied to determine the physical forces that drive the colloids/particles into motion. Indeed, the colloid structure can also be varied so it does not respond to the concentration gradient.

If the authors disagree with my perspective (it seems likely), I would encourage them to consider placing elements of the text in their letter of response into the manuscript. Some select aspects are already in the Discussion, but I think a fuller discussion of the distinction (acknowledging that gradients in concentration are a form of information, or pointing out how my perspective it wrong) would be helpful. This will provide the reader with the authors' understanding of how their system differs from the situation where organization is driven by information in the form of chemical gradients/structures of colloids.

REVIEWERS' COMMENTS:

Reviewer #2 (Remarks to the Author):

I have read the authors' thoughtful response to my comments. Overall, however, I am yet to be convinced by their discussion claiming a fundamental distinction between colloids undergoing directed motions in a chemical gradient and the computer-driven motion of their particles based the locations of other particles (I do see a difference in terms of versatility). Specifically, a colloid can be engineered to respond to a chemical gradient in a manner that is dependent on the specified structure of the colloid. I don't see this type of engineered response being distinct from the engineered computer-driven response based on the positions of other particles. In the latter case, the computer has a structure that dictates the outcome, just as the colloid has a structure that dictates the outcome. The structure of both can be varied to determine the physical forces that drive the colloids/particles into motion. Indeed, the colloid structure can also be varied so it does not respond to the concentration gradient.

Thank you very much for this example! We think that in a very general sense one would be able to argue that any interaction (even the ones we call physical) can be described in terms of information flows. The force on a charged particle in an electric field would then map the information of the electric field strength and direction with a particular information processing to the well known Coulomb law. A specific design of the boundary conditions for the physical interactions could change the response or effectively the information processing. There seem to be theoretical efforts for such a generalised information based description of the physical world, which becomes even more relevant in the world of quantum information.

In what we have tried to lay out, we have made use of a definition which separates the contributions from the fundamental physical interactions from the contributions coming from pure information much like the free energy is split up into contributions of internal energy and entropy. We agree that a pure physical response due to, for example, a Coulomb force or a more complex engineered system may be seen as the response to an information about the field, but even in the absence of the physical interaction, there might be an extra contribution coming from the information about the system alone. This latter point is what we tried to highlight with our experiments.

If the authors disagree with my perspective (it seems likely), I would encourage them to consider placing elements of the text in their letter of response into the manuscript. Some select aspects are already in the Discussion, but I think a fuller discussion of the distinction (acknowledging that gradients in concentration are a form of information, or pointing out how my perspective it wrong) would be helpful. This will provide the reader with the authors' understanding of how their system differs from the situation where organization is driven by information in the form of chemical gradients/structures of colloids.

We have added a more wider discussion (hopefully also reflecting the reviewers point of view) to the manuscript.